

# Fluctuation based interpretable analysis scheme for quantum many-body snapshots

Henning Schlömer[1,2,3,4⋆] and Annabelle Bohrdt[2,3,5,6]

**1** Department of Physics and Arnold Sommerfeld Center for Theoretical Physics (ASC),
Ludwig-Maximilians-Universität München, München D-80333, Germany
**2** Munich Center for Quantum Science and Technology (MCQST),
D-80799 München, Germany
**3** ITAMP, Harvard-Smithsonian Center for Astrophysics, Cambridge, MA, USA
**4** Department of Physics and Astronomy, Rice University, Houston, Texas 77005, USA
**5** Department of Physics, Harvard University, Cambridge, Massachusetts 02138, USA
**6** Institut für Theoretische Physik, Universität Regensburg, D-93035 Regensburg, Germany

⋆ H.Schloemer@physik.uni-muenchen.de

## Abstract

Microscopically understanding and classifying phases of matter is at the heart of strongly-correlated quantum physics. With quantum simulations, genuine projective measurements (snapshots) of the many-body state can be taken, which include the full information of correlations in the system. The rise of deep neural networks has made it possible to routinely solve abstract processing and classification tasks of large datasets, which can act as a guiding hand for quantum data analysis. However, though proven to be successful in differentiating between different phases of matter, conventional neural networks mostly lack interpretability on a physical footing. Here, we combine confusion learning [1] with correlation convolutional neural networks [2], which yields fully interpretable phase detection in terms of correlation functions. In particular, we study thermodynamic properties of the 2D Heisenberg model, whereby the trained network is shown to pick up qualitative changes in the snapshots above and below a characteristic temperature where magnetic correlations become significantly long-range. We identify the full counting statistics of nearest neighbor spin correlations as the most important quantity for the decision process of the neural network, which go beyond averages of local observables. With access to the fluctuations of second-order correlations – which indirectly include contributions from higher order, long-range correlations – the network is able to detect changes of the specific heat and spin susceptibility, the latter being in analogy to magnetic properties of the pseudogap phase in high-temperature superconductors [3]. By combining the confusion learning scheme with transformer neural networks, our work opens new directions in interpretable quantum image processing being sensible to long-range order.

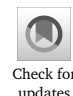

# Contents

# 1 Introduction

Next to revolutionizing applications in image and sequence processing, in recent years neural networks have gained tremendous interest also in the field of quantum many-body physics [4–7]. In strongly correlated systems, complex phases of matter can emerge in seemingly simple models – which, in many settings, still lack microscopic understanding [8,9]. With their powerful abstraction tools, neural networks have quickly opened a novel paradigm of analyzing many-body phases of matter, which may help to gain deeper understanding of appearing phases in strongly correlated systems [2,10,11], as well as act toward experimental image reconstruction [12], enhanced Monte Carlo sampling [13–16], and efficient representations of quantum states [17–19].

As a concrete example, deep neural networks have been increasingly utilized to predict phase transitions in physical systems, the model's input data types ranging from entanglement entropy spectra [1,20–22] to quantum image data generated numerically [23–28] and experimentally [29–32]. However, one major drawback of the neural network toolbox is their inherent black-box nature, which limits interpretation – and in turn restricts their applicability towards developing microscopic theories of yet unsolved physical regimes. For phase classification tasks using standard feed forward neural networks, for instance, the models represent complicated non-linear functions that are optimized to best represent the conditional probability $P(y|\mathbf{s})$ of assigning phase label $y$ to data input $\mathbf{s}$, however mostly without any deeper insights into the decision making process of the network. This significant pitfall of neural networks in quantum physics has triggered intensified research regarding reliable interpretability, such as for linear and kernel [33–35], shallow [36,37] and engineered [2,38,39] models, as well as by using Hessian based similarity measures [40,41] and optimal prediction methods [42,43].

Highly controllable analog quantum simulation platforms – e.g. via ultracold atoms – allow for a systematic experimental exploration of paradigmatic Hamiltonians with strong correlations like the Fermi-Hubbard model [44–52]. In particular, these setups allow to perform

genuine quantum projective measurements and sample snapshots of the many-body state in the Fock basis, which in turn allow for insights into the wave function beyond averages and local observables [48, 53, 54]. Nonetheless, if order parameters are unknown or the physics goes beyond the Landau paradigm of phase transitions, it is a difficult task to differentiate between different phases of matter when a whole zoo of possible correlation functions needs to be considered.

To this end, neural network processing of quantum snapshots can act as a guiding hand, where architectures are desirable that, apart from detecting qualitatively different physical regimes, let us know which physical correlations are crucial to base a reliable decision on. For this purpose, unsupervised-supervised hybrid machine learning approaches based on correlation convolutional neural networks (CCNN) [2] have been proposed, where interpretable phase detection has been demonstrated via data clustering and subsequent filtering of important correlations in each cluster [11]. In this work, we propose a fully automated, unsupervised method for interpretable phase detection by combining confusion learning training schemes [1] with CCNNs, allowing for a reliable single-step detection of qualitatively differing regimes while at the same time yielding full interpretability in terms of correlation functions.

In particular, we study numerically generated snapshots of the Heisenberg model, whose temperature dependent magnetic properties share many similarities with the low-energy features of the Fermi-Hubbard model at half filling. In this case, a characteristic temperature $T^*$ exists where spin correlations become significantly long-range, replacing Fermi-liquid quasi-particles by a single-particle pseudogap [55, 56]. In the Heisenberg model, though no quasi-particle interpretation exists, a suppression of the spin susceptibility can be observed below a characteristic temperature $T^* \sim J$ [57–59] in analogy to the half filled Fermi-Hubbard model [60, 61]. Similarly, both the Heisenberg and Fermi-Hubbard model feature a maximum of the specific heat at $T_C \sim 2J/3$ [62–64], signaling the thermal activation of the spin degrees of freedom. We show that the trained confusion correlator convolutional neural network is able to pick up upon qualitative changes of these thermodynamic properties in the Heisenberg snapshot data sets above and below a characteristic temperature, broadly matching both the peak of the susceptibility as well as the specific heat. We find that the network classifies snapshots by analyzing the full counting statistics of nearest neighbor spin correlations, which directly contain information about higher moments of the distributions. By evaluating the fluctuations of nearest neighbor correlators, the network uses indirect access to four-point correlations to assess long-range properties of the snapshots. Initiating the step towards fully long-range capabilities, we show that similar features are detected using transformer vision networks that include attention and thus correlations across the entire snapshot.

With its powerful correlation based ability to detect even featureless variations of snapshot data sets, our work paves the way towards deep microscopic insights into strongly correlated phases. In particular, application of the fluctuation based detection scheme promises novel perspectives onto non-local properties of many-body systems.

## 2 Correlation based confusion learning

Confusion learning is a training scheme which aims to identify phase transitions by learning the best labeling of data, where the labeling is originally unknown [1]. Given an input dataset in some parameter space $p \in [p_1, p_2]$, purposely mislabeling the data into two subsets and evaluating the performance of the network to distinguish between the two labels can give insights into whether and where a phase transition occurs. Concretely, consider a physical system with a phase transition at point $p_c$. Within the confusion learning scheme, a neural network is trained to distinguish whether the input is taken from $p_1 \leq p \leq p'$ or $p' < p \leq p_2$,

with $p'$ an arbitrary decision boundary. If we choose, for instance, $p' = p_2$, we train the model to assign label "A" to the full dataset, which is a trivial task for the neural network and results in 100% accuracy. The same argument holds if we choose $p' < p_1$, where now all inputs are predicted to belong to label "B". Furthermore, assuming the model is capable of perfectly distinguishing the two phases, we reach ideal performance also at $p' = p_c$. In between, the model is assigned to label data from the same phase as coming from qualitatively different regimes, leading to a majority decision and a reduced accuracy (hence the confusion of the network). As a result, a characteristic W-shape of the network's performance as a function of $p$ is expected.[1] By identifying the maximum of the network's performance $p'_{\max}$ upon varying the decision boundary, the critical parameter $p_c = p'_{\max}$ can be estimated. If, on the other hand, no transition exists in the system, the network will always make a majority decision – resulting instead in a V-shape of the accuracy.

## 2.1 Network architecture

With increasing efforts to interpret machine learning in the context of physical observables, a neural network architecture based on non-linearities that directly correspond to measurable correlations has been proposed in [2]. In particular, the uncontrolled mixing of correlations that appears when using standard non-linearities is explicitly replaced by interpretable correlation maps in the correlation convolutional neural network (CCNN) architecture. Here, by combining correlation based convolutions with confusion learning (co-CCNN), we detect qualitative variations of quantum many-body snapshots while having direct access to the model's decision making process. The network's architecture is schematically illustrated in Fig. 1 (a). Many-body snapshots for a range of parameters are divided into two subsets – i.e., above and below a given decision boundary $p'$. In order to perform interpretable classification, convolutional filters generate a first order correlation map of the snapshot ($C^1$), from which higher order (i.e. non-linear) correlations are evaluated up to order $M$ ($C^n$, $1 < n \leq M$). In particular, for a snapshot with pixels $S_c(\mathbf{x})$ for channels $c = \{\uparrow, \downarrow\}$ and filter weights $f_c(\mathbf{x})$, the correlation maps are given by [2]

$$C^n(\mathbf{x}) = \sum_{(\mathbf{a}_1, c_1) \neq \ldots \neq (\mathbf{a}_n, c_n)} \prod_{j=1}^{n} f_{c_j}(\mathbf{a}_j) S_{c_j}(\mathbf{x} + \mathbf{a}_j), \tag{1}$$

where $\mathbf{a}_j$ refers to the positions of the convolution window.[2] Hence, the $n$'th order correlation map corresponds to $n$-point correlations within a given fixed convolutional window. Note that the above can be easily generalized to multiple filters. However, for the sake of simplicity and easier interpretability, we here restrict ourselves to a single filter per channel.[3] After post-processing the correlation maps by normalizing and averaging,[4] the $M-$dimensional output is fed into a single fully connected layer with weights $w^{(n)}$, which then makes a binary classification based on the measured correlations. As the filters are trainable, the CCNN hence learns which correlations give key information about the two subsets of snapshots when attempting to distinguish between them. Upon sweeping the decision boundary through parameter space, this allows for interpretable classification of snapshots within a single-step protocol in a fully automated manner, whereby the model outputs regions of qualitatively differing snapshot sets

---

[1]Note that, in most realistic applications, the model is not perfectly able to distinguish between the two phases, leading to a smeared out W-shape in the accuracy [1].

[2]$C^1$ thus corresponds to the feature map of a standard convolutional operation; the non-linear part of the model corresponds to all higher orders, $C^n$, $n > 1$.

[3]When including multiple filters, our findings do not change qualitatively.

[4]We assume translational invariance of the system, such that we can get meaningful quantities (i.e. measurable $n$-point correlations) by spatially averaging over the correlation maps.

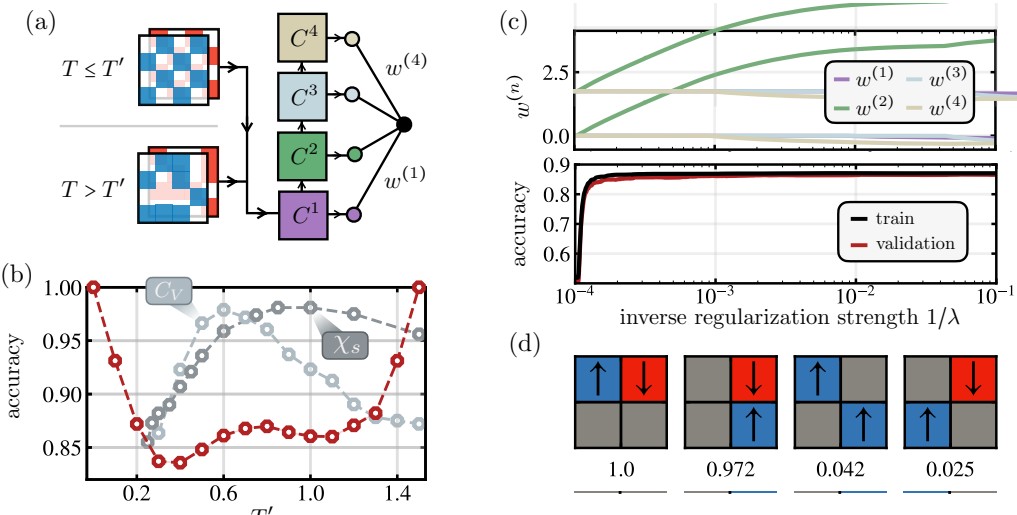

Figure 1: **Correlator based confusion learning.** (a) Schematic architecture of the confusion based [1] CCNN [2] network (co-CCNN). Correlation maps are computed via convolution with learnable filters, which a coupled fully-connected discriminator bases its binary decision on. We use a single $2 \times 2$ filter for each channel, and cascade the first order correlation map to fourth order, $M = 4$. (b) When changing the decision boundary $T'$ in the confusion learning scheme, the network's accuracy features a W-shape, signaling that two qualitatively differing regimes are present in the data (red data points). Accuracies are averaged over 20 runs; errors are negligible on the scale of the plot. The performance maximum at $T'_{\mathrm{max}} \sim 0.8$ is found to broadly match peaks of the specific heat $C_V$ at $T_C \sim 0.6$ (light blue data points) and magnetic susceptibility $\chi_s$ at $T^* \sim 0.9$ (light blue data points, taken from [57]). Values for $\chi_s$ and $C_V$ are re-scaled and shifted to match the axis frame. (c) Top panel: regularization path analysis of the weights $w^{(n)}$ at $T' = 0.7$. Second order correlations are found to set in earliest, while all other correlations stay insignificant for the whole range of $\lambda$. Lower panel: accuracy of the discriminator upon tuning the regularization strength $\lambda$. When significant weight is on the second order correlation neuron, maximum accuracy is reached. (d) The four most relevant two-point correlations that the network utilizes to make its decision, with weights given by $f_{c_1}(\mathbf{a}_1) f_{c_2}(\mathbf{a}_2)$ (normalized by the highest value), cf. Eq. (1). Nearest neighbor correlators are seen to single out as the important correlations.

while at the same time yielding insights into which correlations are important to distinguish these sets.

## 2.2 Application to the Heisenberg model

Using stochastic series expansion quantum Monte Carlo techniques [47,65,66], we take snapshots of the antiferromagnetic (AFM) Heisenberg model at temperature $T$, described by the Hamiltonian [67,68]

$$\mathcal{H} = J \sum_{\langle i,j \rangle} \hat{\mathbf{S}}_i \cdot \hat{\mathbf{S}}_j \, , \tag{2}$$

where $\hat{\mathbf{S}}$ is a spin-1/2 operator and $\langle i,j \rangle$ denotes nearest neighbor pairs on the 2D square lattice. Though long-range antiferromagnetic (AFM) order is only present in the ground state ($T = 0$) and there exists no phase transition at finite temperature, the 2D Heisenberg model features interesting thermodynamic properties. For instance, a typical temperature scale $T^*$

exists at which magnetic correlations become significantly long-range, indicated by a sudden suppression of the uniform spin susceptibility [57–59],

$$\chi_s = \frac{1}{NT} \sum_{i,j} \langle \hat{S}_i^z \hat{S}_j^z \rangle \,, \tag{3}$$

where $N$ the number of spins in the system. The Heisenberg model is an effective low energy description of the Fermi-Hubbard model at half filling and strong repulsion, where a similar phenomenology of the spin susceptibility is observed [55,60]. Here, it has been proposed that at $T^*$, the exponentially growing correlation length of spin fluctuations becomes comparable to the quasiparticle de Broglie wavelength $\lambda_B \sim v_F/T$ (with $v_F$ the Fermi velocity) – leading to the formation of precursor AFM bands and the depletion of the electronic density of states at the Fermi level known as the pseudogap [55]. Early experimental findings of cuprate high-temperature superconductors have established the existence of a pseudogap in doped Mott insulators, however a universal understanding of its origin and in particular its relation to superconductivity is yet to be established [69]. Though subtle differences between the actual opening of the pseudogap at the Fermi surface in the Fermi-Hubbard model and the peak of the magnetic susceptibility exist in cuprate materials [70], $T^*$ – here defined as the maximum of the susceptibility – constitutes a characteristic temperature below which significant magnetic correlations develop.

Moreover, large correlation lengths at low temperatures and random, uncorrelated spins at high temperatures lead to the appearance of a maximum of the specific heat at $T_C \sim 2J/3$ both in the Heisenberg as well as Fermi-Hubbard model,

$$C_V = \left( \langle \hat{\mathcal{H}}^2 \rangle - \langle \hat{\mathcal{H}} \rangle^2 \right)/T^2 = \frac{\partial}{\partial T} \langle \hat{\mathcal{H}} \rangle \,, \tag{4}$$

constituting a characteristic energy scale where spin degrees of freedom are thermally activated [62,64]. At low temperature, it has been shown that $C_V \propto T^2$ [63], as anticipated from spin-wave theory.

The close correspondence of the low energy physics between the Heisenberg and the Hubbard model at half filling together with its non-trivial thermodynamic behavior render the Heisenberg model a valuable and, importantly, verifiable testing ground for machine learning applications. In the following, we analyze simulated snapshots of the Heisenberg model at various temperatures using the co-CCNN scheme. We demonstrate that the network is capable of picking up qualitative thermodynamic changes of the model, which we fully interpret in terms of full counting statistics of correlation functions – paving the way to analyze many-body snapshots in, e.g., temperature and density scans in the Fermi-Hubbard model away from half filling.

In our simulations, we take snapshots of the thermal ensemble of a $40 \times 40$ Heisenberg model, but use only the central $16 \times 16$ region for further processing to minimize boundary effects. In the following, all energies are given in units of $J$, where we set $J = 1$. According to the scheme outlined in Sec. 2.1, we train a CCNN to discriminate between temperatures $T \leq T'$ and $T > T'$ using binary cross entropy (BCE) loss and $2 \times 2$ convolutional filters. We use 2,000 snapshots for each temperature value between $T = 0.1$ and $T = 1.5$ in increments of $\Delta T = 0.1$. We utilize 90% of the data set for training; the remaining 10% is used for validation. The accuracy after 50 training epochs averaged over 20 runs is shown in Fig. 1 (b). In immediate vicinity to the boundaries $T' \gtrsim 0.1$ and $T' \lesssim 1.5$, we see a linear reduction of accuracy, signaling that the network makes a majority decision. At intermediate decision boundaries, however, the network's performance reaches a local maximum located at $T'_{\max} \sim 0.8$ – being in broad agreement with both the maximum of the spin susceptibility at $T^* \sim 0.9$ (dark grey data points in Fig. 1 (b)) as well as the peak of the specific heat at $T_C \sim 0.6$ (light grey data points in

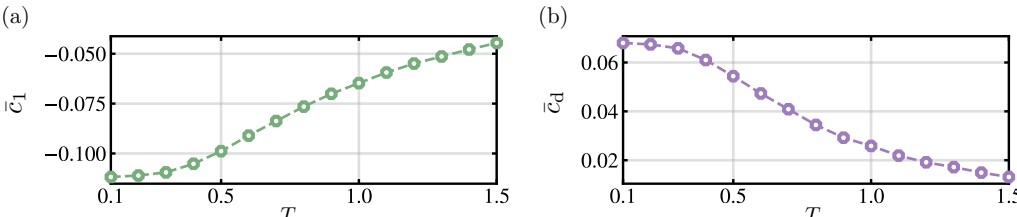

Figure 2: **Two-point correlations of the Heisenberg model.** Nearest neighbor correlations $\bar{c}_1$, (a), as well as diagonal correlations $\bar{c}_\mathrm{d}$, (b), in the Heisenberg model as a function of temperature. Correlations are approximated by evaluating $c_1, c_\mathrm{d}$ in each shot and averaging over all snapshots, cf. Eq. (7). Both correlations show a monotonous behavior, with no qualitative differences above and below $T'_\mathrm{max} \sim 0.8$.

Fig. 1 (b), evaluated by numerical differentiation of $\langle \hat{\mathcal{H}} \rangle$. As we show in the Appendix, Sec. A, our results to not alter qualitatively when choosing larger convolutional windows. However, generally, the filter size shall be treated as a tunable hyperparameter of the CCNN, whereby the maximum order of correlations accessible to the model is limited by the size of the kernel.

The observed performance maximum at $T'_\mathrm{max}$ suggests that the network picks up upon the qualitative change of thermodynamic properties of the spin system below and above characteristic energy scales $T_C, T^*$, where magnetic correlations become significantly long-range. Importantly, we note that quantities such as $C_V$ and $\chi_s$ explicitly include long-range contributions, cf. Eqs. (3), (4); the network, however, is restricted to evaluating local correlations within the convolutional filter. Thus, the question arises how the model makes its decision and which qualitative changes precisely the co-CCNN scheme detects.

### 2.2.1 Regularization path analysis

To classify which correlation maps are important for the decision process, we retrain the fully connected layer of the model that directly leads to the decision neuron [2]. By explicitly adding a $\mathcal{L}_1$ penalty to the weights $w^{(n)}$ between the post-processed correlation maps and the interpretation bottleneck (see Fig. 1 (a)), we perform a regularization path that allows us to analyze which correlation map is used first to reach maximum discrimination accuracies. In particular, the retraining loss reads

$$\mathcal{L}_\mathrm{reg} = \mathcal{L}_\mathrm{BCE} + \lambda \sum_{n=1}^{N} \left| w^{(n)} \right| , \tag{5}$$

where $\mathcal{L}_\mathrm{BCE}$ is the binary cross entropy loss that was used to train the convolutional filters and $\lambda$ is the regularization strength.

Weights $w^{(n)}$ for a given $\lambda$ are shown in the top panel of Fig. 1 (c) for decision boundary $T' = 0.7$. We observe that for $1/\lambda \sim 10^{-4}$, weights for the second order correlations first start to deviate from zero. At the same time, the accuracy of the network shoots from $\sim 50\%$ to $\sim 85\%$, see the lower panel in Fig. 1 (c). Note that all other weights are vanishingly small throughout the whole range of $\lambda$, and even when slightly deviating from zero do not lead to a performance increase of the network. Hence, we conclude that it is indeed correlations of second order that let the network reach its maximal accuracy shown in Fig. 1 (b).

To make it explicit which two-point correlators precisely the network measures, we plot the four correlations with highest weights $f_{c_1}(\mathbf{a}_1) f_{c_2}(\mathbf{a}_2)$ (normalized by the largest correlation weight) when applying the learned convolutional filter, Fig. 1 (d). Nearest neighbor spin-spin correlations in horizontal and vertical direction single out by their strong weights. Diagonal correlations are found to be further calculated and analyzed by the network, however only

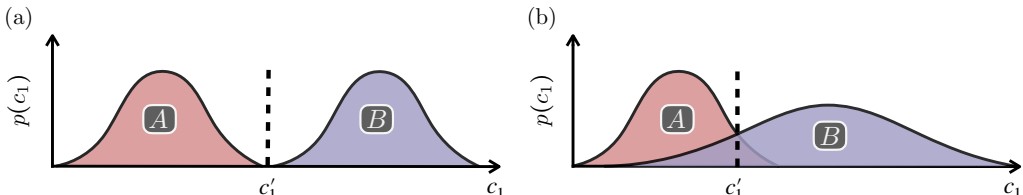

Figure 3: **Illustration of the network's learning process.** By analyzing the full counting statistics of sets $A = \{c_1 \,|\, T \leq T'\}$ and $B = \{c_1 \,|\, T > T'\}$, the network learns a threshold $c_1'$. For a given, unseen snapshot with $c_1^{(s)}$, the network then classifies it as belonging to $T \leq T'$ ($T > T'$) for $c_1 \leq c_1'$ ($c_1 > c_1'$). When both distributions have no overlap, the network has perfect accuracy, (a); for finite overlap, the network's performance decreases, (b). The ideal choice of $c_1'$ that maximizes the accuracy explicitly depends on the full distributions of $A$ and $B$, including their means and widths.

with marginal weight (around 5%) compared to the nearest neighbor correlations. Note that for all decision boundaries $T'$, the results shown above are qualitatively identical – that is, second order nearest-neighbor correlations are found to be used by the network to make its decision.

### 2.2.2 Full counting statistics

Based on these insights, we take a look at nearest neighbor and diagonal spin-spin correlations,

$$\langle \hat{c}_1 \rangle = \left\langle \frac{1}{N_b} \sum_{\langle i,j \rangle} \hat{S}_i^z \hat{S}_j^z \right\rangle, \qquad \langle \hat{c}_d \rangle = \left\langle \frac{1}{N_b} \sum_{\langle\langle i,j \rangle\rangle} \hat{S}_i^z \hat{S}_j^z \right\rangle, \tag{6}$$

where $N_b$ is the total number of nearest neighbor (diagonal) pairs $\langle i,j \rangle$ ($\langle\langle i,j \rangle\rangle$). We evaluate correlations Eq. (6) by averaging over $N_s$ snapshots,

$$\bar{c}_1 = \frac{1}{N_s} \sum_{s=1}^{N_s} c_1^{(s)}, \qquad \bar{c}_d = \frac{1}{N_s} \sum_{s=1}^{N_s} c_d^{(s)}, \tag{7}$$

where $c_{1/d}^{(s)}$ is the approximation of the correlator $c_{1/d}$ using snapshot $s$,

$$c_1^{(s)} = \frac{1}{N_b} \sum_{\langle i,j \rangle} S_i^{z,s} S_j^{z,s}, \qquad c_d^{(s)} = \frac{1}{N_b} \sum_{\langle\langle i,j \rangle\rangle} S_i^{z,s} S_j^{z,s}, \tag{8}$$

with $S_i^{z,s}$ denoting the spin orientation of spin $i$ in snapshot $s$. As depicted in Fig. 2, both correlator strengths show a monotonous increase with decreasing temperature with no qualitative difference above or below the temperature of maximum network accuracy.

If no structural change in the two-point correlators can be seen when passing $T'_{\max}$, but the network only utilizes nearest neighbor two-point correlations when learning to label the data, what is it then that the network bases its decision upon?

To answer this question, we analyze the full counting statistics (FCS) of $c_1$, given by the total distribution $\{c_1\} = \{c_1^{(1)}, c_1^{(2)}, \dots\}$, cf. Eq. (8). In contrast to merely using averages, Eq. (7), the FCS directly gives information about higher moments of the distribution, such as its width and skewness. In particular, given a bipartition of the data set with boundary $T'$, a corresponding boundary $c_1'$ can be learned by the network which assigns label $T \leq T'$ ($T > T'$) to all snapshots fulfilling $c_1^{(s)} \leq c_1'$ ($c_1^{(s)} > c_1'$) such that its accuracy is maximized. If the network indeed estimates $c_1$ for each snapshot and bases its decision on the result, its accuracy will be

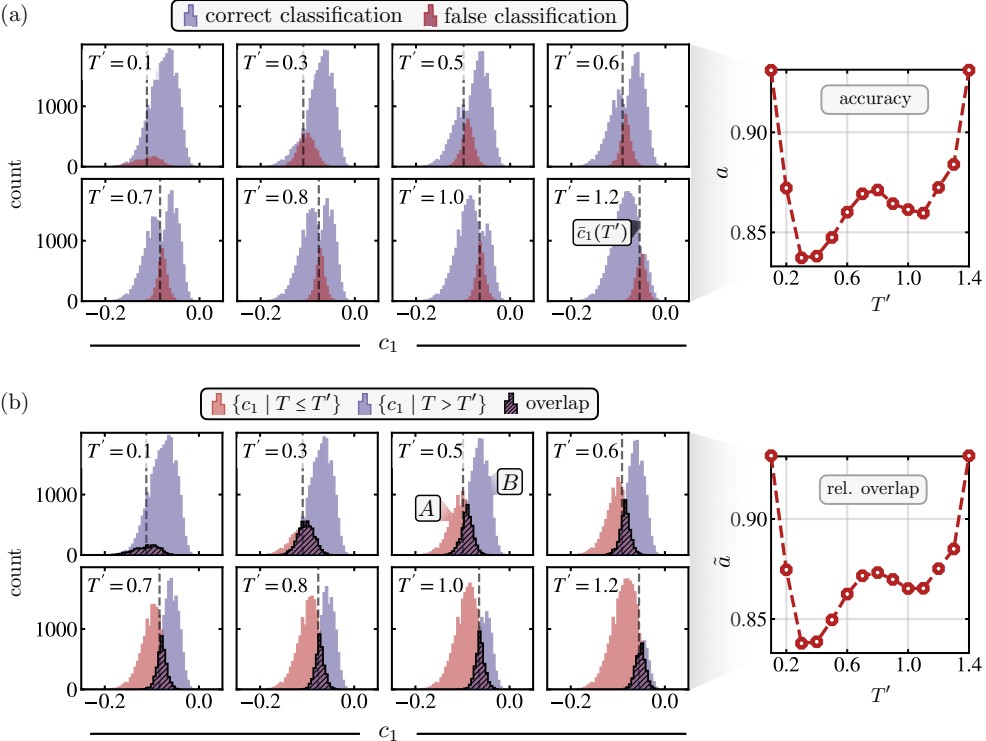

Figure 4: **Full counting analysis.** (a) We approximate the nearest neighbor corre-
lator for each snapshot, Eq. (8), and analyze whether the network correctly labels it
for a given decision boundary $T'$ after training. Shown are the full counting statis-
tics, where blue (red) indicates a correct (wrong) decision by the network. The right
panel shows the total accuracy of the network, cf. Fig. 1 (b). (b) Full counting statis-
tics of $c_1$ directly calculated from the Heisenberg snapshot data for a given bipartition
$T \leq T'$ (red), $T > T'$ (blue). If basing the labeling decision on $c_1$, finite errors are
expected when both distributions overlap (hatched areas). The shapes of the hatched
and non-hatched areas precisely match the FCS of the network's performance as a
function of $c_1$ in (a), underlining the interpretation of the network's decision mak-
ing. The right panel shows the ratio $\tilde{a}$ between the area spanned by the non-hatched
distributions to the total area below both hatched and non-hatched distributions, re-
producing the W-shape of the network's accuracy.

flawless if the two sets $A = \{c_1 \,|\, T \leq T'\}$ and $B = \{c_1 \,|\, T > T'\}$ have no overlap, as illustrated
in Fig. 3 (a). On the other hand, increasing overlaps will result in decreasing accuracy of the
network as a hard decision boundary $c_1'$ will inevitably lead to uncertain label predictions, cf.
Fig. 3 (b).

For each shot, we calculate the snapshot's approximation of $c_1$, Eq. (8), and explicitly differ-
entiate whether or not the network makes a correct decision, $C = \{c_1 \,|\, \text{correct categorization}\}$
and $F = \{c_1 \,|\, \text{false categorization}\}$, shown in Fig. 4 (a) in blue and red, respectively. The ac-
curacy of the classifier is hence given by $a = {|C|}/{|C|+|F|}$, with $|C|$ ($|F|$) referring to the total
instances of correctly (incorrectly) categorized snapshots. Accuracies $a$ as a function of $T'$ are
shown on the right hand side of Fig. 4 (a), matching Fig. 1 (b).[5]

We now perform a similar analysis of the FCS of $c_1$ directly from the raw Heisenberg snap-
shot data. To this end, we create bipartitions $A = \{c_1 \,|\, T \leq T'\}$ and $B = \{c_1 \,|\, T > T'\}$ of the

---

[5]Note that in Fig. 4 (a) we are showing the accuracy after a single run, whereas Fig. 1 (b) presents the mean
accuracy over multiple optimizations – leading to the two curves not to be identical.

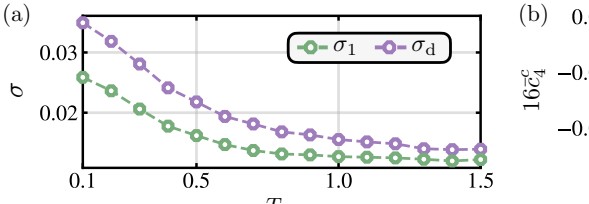
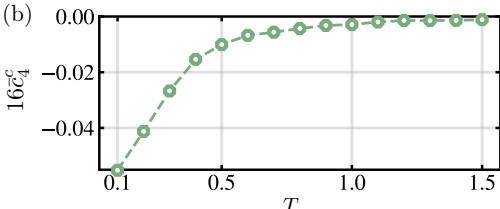

Figure 5: **Standard deviations and connected four-point correlations.** (a) The empirical standard deviation $\sigma_{1/\mathrm{d}}$, Eq. (9), of $c_1, c_\mathrm{d}$, showing a sharp increase below temperature $T'_{\max} \sim 0.8$. (b) Averaged connected four-point correlator, Eq. (11), as a function of $T$. While being vanishingly small for $T \gtrsim T'_{\max}$, for $T \lesssim T'_{\max}$ the connected correlator gains significant weight.

snapshots and plot the corresponding distributions, in analogy to Fig. 3. Results are shown in Fig. 4 (b), where A (B) is shown in red (blue). Overlaps of both distributions are illustrated by hatched areas. Comparing the histograms in Fig. 4 (a) and (b), we find that the distributions match up almost perfectly. In particular, the hatched overlap of $A$ and $B$ in Fig. 4 (b) corresponds to the distribution of false classifications of the network, $F$, see Fig. 4 (a). Correct instances $C$, in turn, match the distribution $(A \backslash B) \cup (B \backslash A)$, i.e., the non-hatched areas in Fig. 4 (b). Indeed, when computing the accuracy analog of the raw Heisenberg histograms by evaluating the ratio $\tilde{a} = {}^{|A \backslash B| + |B|}\!/\!_{|A| + |B|} = {}^{|B \backslash A| + |A|}\!/\!_{|A| + |B|}$, the characteristic W-shape of the confusion learning scheme is reproduced — even matching quantitatively the accuracy of the neural network up to high precision, see the right panel in Fig. 4 (b).

For a given snapshot $s$ to be categorized, we conclude that the network makes a majority decision that is based on the snapshot's nearest neighbor correlation estimate $c_1$. In particular, the network learns a threshold $c'_1$ according to which it classifies a given snapshot with $c_1^{(s)}$ as $T \le T'$ or $T > T'$ for $c_1^{(s)} \le c'_1$ and $c_1^{(s)} > c'_1$, respectively. We note that, as the network has no information about the temperature of the snapshots, it can not, for instance, estimate averages $\bar{c}_1(T')$ and make a corresponding decision according to $c'_1 = \bar{c}_1(T')$. Instead, the network leverages the FCS of the distributions $A$ and $B$, choosing $c'_1$ in order to maximize the classification accuracy. Specifically, $c'_1$ corresponds to the point where the distributions $A = \{c_1 \,|\, T \le T'\}$ and $B = \{c_1 \,|\, T > T'\}$ have maximum overlap, cf. Figs. 3 and 4. The learned threshold $c'_1$ explicitly depends on the widths $\sigma$ of distributions $A$ and $B$, which directly include information of the fluctuations of nearest neighbor correlations $c_1$. We note that the learned decision thresholds closely (though not exactly) match the values of $\bar{c}_1(T')$, as illustrated in Fig. 4 by grey dashed lines. In the Appendix, Sec. A, we show that indeed a sharp cutoff exists at $c'_1$ below (above) which the network categorizes snapshots as $T \le T'$ ($T > T'$) – see Fig. 7.

Having identified the FCS of $c_1$ as the decisive mechanism of the network to detect qualitatively differing snapshots in the Heisenberg model, we take a closer look at the widths $\sigma_1$ of the distributions $\{c_1\}$ as a function of temperature, i.e., we study the fluctuations of $c_1$,

$$\sigma_1^2 = \langle \hat{c}_1^2 \rangle - \langle \hat{c}_1 \rangle^2 \,. \tag{9}$$

Results are shown in Fig. 5 (a). We observe that at high temperatures, the width of the distributions stay relatively constant. At roughly $T \sim T'_{\max} \sim 0.8$, however, the standard deviation starts to significantly increase, consistent with magnetic fluctuations becoming more prominent at temperatures below $T^*$. As shown in Fig. 5 (a), this holds for both nearest neighbor as well as diagonal two-point correlations.

Explicitly writing out Eq. (9),

$$\sigma_1^2 = \frac{1}{N_b^2} \sum_{\langle i, i' \rangle} \sum_{\langle j, j' \rangle} \langle \hat{S}_i^z \hat{S}_{i'}^z \hat{S}_j^z \hat{S}_{j'}^z \rangle - \langle \hat{c}_1 \rangle^2 \,, \tag{10}$$

we see that $\sigma_1$ in fact includes four-point correlations over two nearest neighbor spin pairs – which, for a given configuration of indices $\langle i, i' \rangle$, $\langle j, j' \rangle$ might lie far apart from each other. Hence, the width of the distribution of $c_1$ directly includes information about long-range properties of the spin-spin correlations. Note that, while nearest neighbor two-point correlations show monotonous behavior as a function of temperature, long-range two-point correlations as appearing in the susceptibility, Eq. (3), *do* show signals of changes of the thermodynamic properties, cf. Fig. 1 (b). However, as the network is by construction restricted to analyze local correlations only, it has no access to evaluate these long range properties. By instead considering the FCS of $c_1$, the network finds a back door to analyze long-range correlations via the four-point correlator appearing in Eq. (10), which enables it to detect qualitatively different thermodynamic characteristics of snapshots above and below $T'_{\max}$.

In fact, the width of $c_1$, Eq. (9), very closely resembles the form of the specific heat $C_V$, Eq. (4). Concretely, $\sigma_1^2$ corresponds to the Ising part of $T^2 C_V$, where in particular cross-terms such as $\sim \langle \hat{S}_i^x \hat{S}_{i'}^x \hat{S}_j^z \hat{S}_{j'}^z \rangle$ as appearing in $C_V$ are not present. Though there exists no pronounced peak of $\sigma_1$ as observed for the specific heat at $T_C \sim 0.6$, its strong alternation for $T \lesssim 0.8$ is highly suggestive of corresponding thermodynamic features appearing in $C_V$, cf. Fig. 1 (b). However, though similarities are present, there exists no direct correspondence between the FCS signatures the network utilizes and thermodynamic properties such as $C_V$ or $\chi_s$. By indirectly evaluating long-range properties (as appearing in $\chi_s$ and $C_V$) of four-point correlations (as appearing in $C_V$), the network succeeds in detecting qualitative changes in the snapshots as a function of temperature. These detected changes cannot, however, directly be attributed to originating from the peak in $C_V$ or $\chi_s$, and shall rather be interpreted as a related but independent indicator of qualitative change close to the pseudogap regime – as also suggested by the position of the performance maximum lying in between the peaks of $C_V$ and $\chi_s$. Nevertheless, the presence of pronounced signatures of $\sigma_1$ as a function of temperature is very intriguing by itself, in turn strongly encouraging the observation of similar indications at finite doping in spin-resolved occupation number snapshots as accessed through quantum gas microscopes.

We conclude the above discussion by calculating explicitly the connected four-point spin correlator,

$$\langle \hat{c}_4^c \rangle = \frac{1}{N_b^2} \sum_{\langle i, i' \rangle} \sum_{\langle j, j' \rangle} \Big[ \langle \hat{S}_i^z \hat{S}_{i'}^z \hat{S}_j^z \hat{S}_{j'}^z \rangle - \langle \hat{S}_i^z \hat{S}_{i'}^z \rangle \langle \hat{S}_j^z \hat{S}_{j'}^z \rangle - \langle \hat{S}_i^z \hat{S}_j^z \rangle \langle \hat{S}_{i'}^z \hat{S}_{j'}^z \rangle - \langle \hat{S}_i^z \hat{S}_{j'}^z \rangle \langle \hat{S}_{i'}^z \hat{S}_j^z \rangle \Big], \quad (11)$$

which we again approximate using the Heisenberg snapshots, $\bar{c}_4^c$. Eq. (11) gives information about the genuine four-point correlations in the system, that in particular go beyond merely the correlation length of the two-point correlators. Evaluation of $\bar{c}_4^c$ shows vanishingly small values for $T > T'_{\max}$, shown in Fig. 5 (b). However, as $T$ drops below $T'_{\max}$, $\bar{c}_4^c(T)$ experiences a sharp increase, indicating how long-range, four-point correlations gain significant weight below $T'_{\max}$ – and correspondingly below the pseudogap temperature, $T^*$, and the maximum of the specific heat, $T_C$.

## 3 Confusion transformer

A general concern when using convolutional neural networks to classify phases as presented above is the limitation of correlations to the convolutional window, which seemingly excludes sensibility to long-range order. As seen above, performant characterization can nevertheless be achieved by the network via analysis of the FCS of local correlations, which implicitly includes longer-range contributions. Nevertheless, a network architecture that is able to intrinsically capture long-range correlations is desirable for future applications of machine vi-

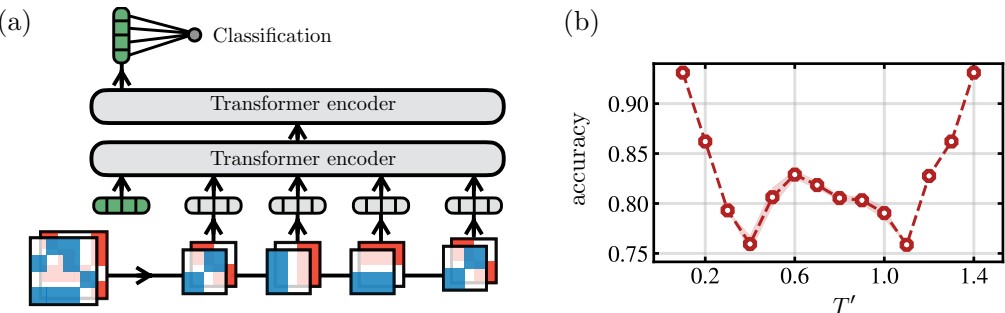

Figure 6: **Confusion transformer.** (a) Schematic illustration of the transformer architecture coupled to a confusion learning scheme. Snapshots are cut into small patches and linearly embedded via learnable matrices. The classification token (shown in green) as well as positional encodings are added to the sequence, before being encoded in two transformer blocks. The classification token – now including information of all patches – is retrieved after the last self-attention encoder and classified to belong to $T \leq T'$ or $T > T'$. (b) Classification accuracy as a function of $T'$. The W-shape signals detection of qualitative change between the regions $T \lesssim 0.6$, $T \gtrsim 0.6$. Light red areas correspond to the error to the mean of 20 repetitions.

sion techniques in many-body physics. Transformers are a promising candidate for this purpose, taking advantage of non-local (and i.p. long-range) self-attention originally designed to capture interdependencies in natural language processing (NLP) [71]. In particular, and in stark contrast to e.g. recurrent neural networks (RNN) and long short-term memory models (LSTM) [72], transformer architectures explicitly avoid recurrent processing of sequential data. Instead, they compute similarity scores between all constituents of a given input sequence (self-attention), allowing to capture long-range dependencies by processing the input as a whole – i.e., they do not rely solely on past hidden states in the sequence.

In the past years, extension of transformers to image processing (vision transformers) has proven itself to be comparably powerful to convolutions [73], opening possible routes toward their application in many-body physics [74–76]. The architecture of a vision transformer is schematically illustrated in Fig. 6 (a). In the first step, input images are cut into smaller patches. These patches are subsequently linearly transformed, i.e., $d-$dimensional representations of the input patches, called tokens, are computed.[6] Importantly, as transformers do not sequentially process the input, the tokens are further positionally encoded, i.e., the position of the patch within the original image is stored. Thereafter follows the self-attention encoder, where all-to-all inter-dependencies between tokens are computed. In particular, three linear transformations are learned, resulting in three feature vectors per token, referred to as query, key and value (QKV). Evaluation of dot-products between query-key pairs results in attention scores between corresponding pairs of tokens, which is then used to efficiently store interdependencies of a given token with the remaining sequence. By feeding the encoded output of a single transformer block into another, independent encoder, this process is repeated multiple times. Additionally, multiple QKV transformations can be learned and applied in parallel in each transformer block, resulting in multi-head attention encodings.

In addition to the data tokens, a randomized classification token is added to the beginning of the sequence, which, via the self-attention mechanism, stores all inter-dependencies between tokens while being processed through the various layers. After application of the encoding blocks, the classification token is passed to a standard classifier. For a more detailed discussion of vision transformer networks, we refer to its original proposal in [73].

---

[6]In NLP, these tokens correspond to encodings of words.

For quantum-image processing, the vision transformer's intrinsic capability of capturing long-range dependencies promises sensitivity to long-range and non-local (e.g. topological) order, potentially leading to significant advantages over standard, convolutional approaches.

We implement a vision transformer and combine it with the same confusion learning scheme outlined in Sec. 2. The original snapshot images are cut into $4 \times 4$ patches, and are fed into the first transformer encoder after a learnable linear embedding and positional encoding[7] is applied. In particular, the $32-$dimensional sequences (16 entries for each channel) are embedded into an $8-$dimensional space (tokens), and a classification token is inserted at the beginning of the sequence (green box in Fig. 6 (a)). For simplicity, we use a single head within the encoder, and limit the network to two transformer blocks. After applying self-attention and classifying the auxiliary classification token, accuracies after training are presented in Fig. 6. Similarly to using convolutional neural networks, a clear W-shape is visible in the accuracy as a function of decision boundary $T'$, with a pronounced maximum at $T'_{\max} \sim 0.6$ – suggesting that also the vision transformer detects the alternations of thermodynamic properties. Through accessing the model's learned attention maps between various patches, their inter-dependencies can be analyzed. In particular, tailored encoding blocks could allow for interpretation in terms of correlation functions similar in spirit to CCNNs, whereby the order of encoded correlations increases with each encoding layer in the transformer architecture. Importantly, the all-to-all self-attention mechanism could surpass convolution based approaches, in particular when facing systems characterized by long-range and non-local properties – which we will look further into in future work.

## 4 Discussion

In this article, we have proposed the co-CCNN scheme as an approach based on interpretable neural networks to detect distinct regimes in quantum many-body snapshots. Specifically, by utilizing correlation-based convolutions in conjunction with a confusion learning scheme, it is possible to identify parameter regions that exhibit significant differences, while maintaining complete interpretability through correlation functions.

We have applied the method to snapshots of a 2D Heisenberg spin system, where the build up of magnetic correlations as the temperature is decreased leads to the appearance of, e.g., pronounced peaks of the specific heat and spin susceptibility. Using our method, we found that the network categorizes the snapshots into two regimes $T \lessgtr T'_{\max}$, whereby $T'_{\max}$ was found to broadly match temperatures of maximal specific heat and susceptibility – thus capturing the variation of thermodynamic properties. We found that the network bases its decision solely on nearest neighbor correlations, which by itself have featureless, monotonic characteristics as a function of temperature. However, we presented strong evidence that the network indirectly accesses long-range, four-point correlations in the system by analyzing the full counting statistics of nearest neighbor correlations. This enables the network to detect alternations of thermodynamic quantities, such as the peak of the specific heat and suppression of spin susceptibility, which directly include contributions from long-range correlations.

With even subtle alternations being detected by the network, this opens up insightful future directions in interpretable quantum image processing. With regard to analog simulation of strongly correlated systems with quantum gas microscopes, the presented method can be directly applied to detect regions of differing phases, with immediate access to correlation functions which are important to characterize the respective regimes. Applying the method to the doped Fermi-Hubbard model might help to pin down the microscopic origin of, for instance, the pseudogap phase, in particular regarding the debated question whether pairing or

---

[7]We use the same positional encoding as proposed in [71].

magnetic fluctuations ultimately lead to the opening of the single particle spectral gap. Concretely, our work suggests to directly look for the four-point spin correlator identified here, both as a function of doping and as a function of temperature at finite doping.

We note that for our numerical experiments, which consist of data sets similar in size to realistic quantum gas microscope experiments, the confusion learning scheme demands only low computational resource. Nevertheless, for larger data sets, the retraining for all possible decision boundaries can quickly become expensive, for which extended schemes as proposed in [77] combined with interpretable CCNN architectures pose a possible extension of our work.

Making the bridge towards networks that have intrinsic capabilities of capturing long-range dependencies, we found that vision transformers trained according to the confusion learning scheme further support the categorization $T \lesssim T'_{\max}$ – promising novel aspects of interpretable machine learning applications in many-body physics.

## Acknowledgments

We thank F. Grusdt, E. Khatami, E.-A. Kim, M. Knap, H. Lange and C. Miles for valuable discussions. We thank M. Kanász-Nagy for providing the Quantum Monte Carlo code. This research was funded by the Deutsche Forschungsgemeinschaft (DFG, German Research Foundation) under Germany's Excellence Strategy—EXC-2111—390 814868, by the European Research Council (ERC) under the European Union's Horizon 2020 research and innovation programme (grant agreement number 948141) – ERC Starting Grant SimUcQuam, and by the NSF through a grant for the Institute for Theoretical Atomic, Molecular, and Optical Physics at Harvard University and the Smithsonian Astrophysical Observatory.

## A  Classification boundary $c'_1$

In the main text, we have seen that the confusion learning trained correlation based convolutional neural network classifies snapshots as belonging to subset $T \leq T'$ or $T > T'$ for a given decision boundary $T'$ by estimating the nearest neighbor correlator $c_1$. To underline the network's decisive mechanism, we compute $c_1$ for each snapshot and create two corresponding subsets by distinguishing between the two classification outcomes by the network after training. Fig. 7 shows the distributions of $c_1$ when being classified as $T \leq T'$ (red) and $T > T'$ (blue) for $T' = 0.1 \dots 1.2$. For $T'$ close to the lower boundary of simulated temperatures, $T' = 0.1$, we see how the network classifies (almost) all snapshots as $T > T'$, hence locking in on a majority decision. However, for intermediate temperatures $0.3 \lesssim T' \lesssim 1.2$, a sharp cutoff between samples classified as $T \leq T'$ and $T > T'$ in terms of $c_1$ is observed. Indeed, the cutoff matches quantitatively the averages $\bar{c}_1(T')$, underlining that the network makes its decision solely by comparing $c_1^{(s)}$ with (a learned) cutoff value given by $c'_1 = \bar{c}_1(T')$.

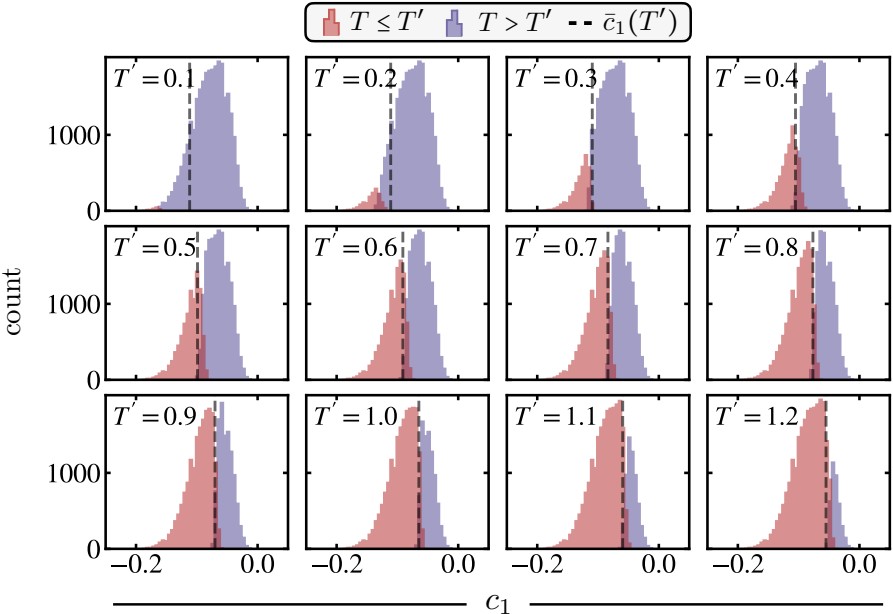

Figure 7: **Classification boundary $c_1'$.** After training, $c_1$ is calculated and sorted into two sets corresponding to their classification. Approximations $c_1$ of snapshots classified as $T \leq T'$ ($T > T'$) are shown in red (blue) for $T' = 0.1 \ldots 1.2$. For $T' = 0.1$, (almost) all snapshots are classified to belong to $T > T'$. For $T' \geq 0.3$, maximum accuracy is instead achieved by learning a boundary $c_1'$, below (above) which all snapshots are classified as $T \leq T'$ ($T > T'$) – underlining that the decisive process of the network is solely based on evaluation of $c_1$. Though not exactly, these cutoffs match averages of $c_1$ at the decision boundary temperature, i.e., $\bar{c}_1(T')$ (grey dashed lines).

## B   Larger convolutional windows

In the main text, we have focused on fixed convolutional filter sizes of $2 \times 2$ and demonstrated that the FCS of two-point correlations enable the network to classify snapshots. We now re-train the model with a single $3 \times 3$ filter, and again analyze the network's performance and regularization path; results are illustrated in Fig. 8. Though showing slight deviations in the network's accuracy as a function of $T'$ from $2 \times 2$ filters, the qualitative W-shape including the position of $T'_{\max}$ remains unchanged when considering larger filter sizes, Fig. 8 (a). As for the $2 \times 2$ filter, including solely two-point correlations leads to maximum accuracy as a function of regularization strength $\lambda$, see Fig. 8 (b). Note that, with increasing inverse regularization strength $1/\lambda$, weights corresponding to higher order correlations also light up, however without any noticeable effect on the network's performance. This highlights the importance of the regularization strength analysis, whereby solely looking at weights of the last dense layer for $\lambda = 0$ is generally not sufficient to reliably learn which correlations are important. In Fig. 8 (c), we show the four two-point correlations with highest weights (corresponding to $f_{c_1}(\mathbf{a}_1)f_{c_2}(\mathbf{a}_2)$, normalized by the highest value). As for the $2 \times 2$ filter, nearest neighbor spin-spin correlations single out as the most important contributions.

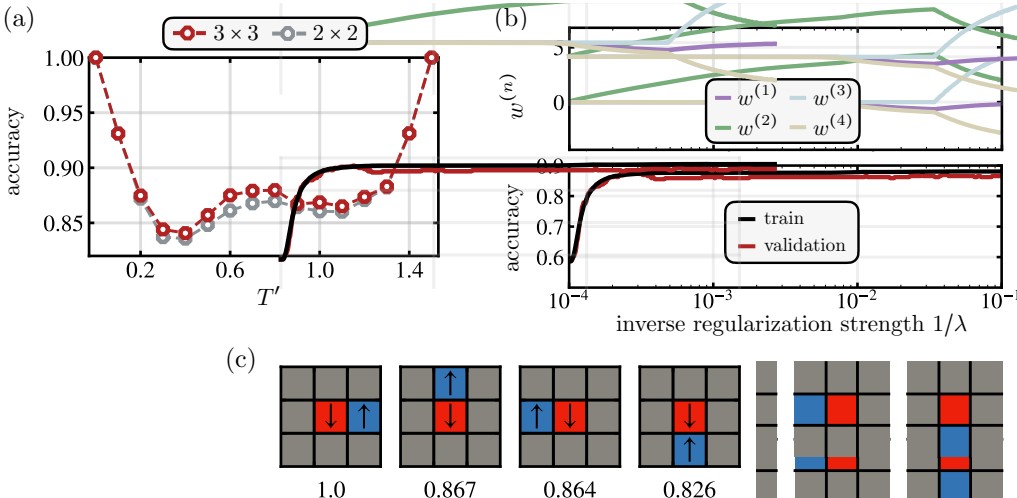

Figure 8: **Larger filter sizes.** (a) The network's performance as a function of decision boundary $T'$ for $3 \times 3$ (red) as well as $2 \times 2$ filters as presented in the main text (grey). Though slight quantitative differences are present, the qualitative shape including the position of the maximum remains unchanged. (b) Regularization path analysis for $3 \times 3$ convolutional filters. Inclusion of two-point correlations lead to a saturation of the accuracy. Finite weights of higher-order correlations as present at large $1/\lambda$ have no effect on the performance on the network, highlighting the importance of the regularization path analysis to isolate the most important contributions. (c) Two-point correlations of highest weights $f_{c_1}(\mathbf{a}_1) f_{c_2}(\mathbf{a}_2)$, normalized by the maximum value. As for the $2 \times 2$ filter, nearest neighbor correlations single out as the most important contributions for the network's decision.

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
