# Peer review of "Fluctuation based interpretable analysis scheme for quantum many-body snapshots"

_SciPost Physics, doi:SciPost Phys. 15, 099 (2023)_

## Round 1 · Referee Report · Anonymous · 2023-5-9

Strengths
1) The work concerns a very interesting "hot" research topic
2) The numerics performed by the authors seems reliable
3) The authors address very carefully the issue of understanding the mechanisms underlying the goodness of the network predictions. This effort is particularly valuable since in many other works Neural Networks are used as a black box, without any physical interpretation of the learned operating mechanism.
Weaknesses
1) I think that the paper is not clear enough in some relevant points (see list below), especially concerning the general features and improvements given by their technique, the possibility of using the approach for other physical models and the use of the "transformer" architecture.
2) Overall, a very clear/innovative/unambiguous take home message is not clearly recognizable.
Report
The paper "Fluctuation based interpretable analysis scheme for quantum many-body snapshots" by Henning Schloemer and Annabelle Bohrdt is a valuable work concerning an extremely relevant field of research, namely the analysis and interpretation of the quantum snapshots acquired via projective measurements from a correlated quantum state.
In this paper, the snapshots are analyzed by means of a Correlation Convolution Neural Network, within the framework of Confusion learning. Although these two tools are already known in litterature, the idea of combining them together is original and interesting, leading to an interpretable tool. The authors discuss one main application of their technique, namely the 2D Heisenberg model, for which they argue to be able to identify a qualitative change of the thermodynamic properties at temperature T*~0.9J, corresponding to a maximum of the susceptibility. The Authors provide a very detailed interpretation of the learned operating mechanism of the Neural Network.
Considering the Strengths/Weaknesses I mentioned and the questions below, I think the work can be published upon some considerable adjustements.
Requested changes
Here are some questions and suggestions for the Authors to consider:
1) A very generic question about the confusion learning approach: under which conditions one can be confident that "the model is capable of perfectly distinguishing the two phases", as you claim for the case $p'=p_c$ at page 4? As far as I understand, this is crucial in order to observe a W-shape in the accuracy as a function of $p'$ (or $T'$).
2) In Section 2.1 you are using $N$ to indicate the maximum order of correlations considered by the network, whereas in Equation 3 $N$ represents the systems size. Maybe you can use a different notation to avoid a misunderstanding.
3) For which reasons are you focusing only on the 2D Heisenberg model? It is just one of the possible applications, or there is a particular motivation for considering it? As far as I understand, you are interested in establishing if the confusion learning approach can detect a qualitative change in the thermodynamical properties (as the one occurring at $T=T^*$), although it is not associated with a proper phase transition. Right? Or rather the reason of interest is given by the link with the low energy physics of the Fermi Hubbard model?
4) I believe that before considering one particular application (i.e. before of Sec. 2.2), you should clarify which are the key features of your technique and in which aspects it leads to an improvement with respect to already known techniques.
5) Some other relevant references on the 2D Heisenberg model shall be added, in order to assist a reader that is not fully familiar with the model and its equilibrium properties. Ref.[55],[60] are good, but probably having some more recent works/reviews/books would help...
6) The agreement between $T_{max}'$~0.8 and $ T^*$~0.9 is not too good. Is there any way of improving the result? Do you expect to find $T_{max}'$ closer to the peak of specific heat or to the the peak of the susceptibility (i.e. $T^*$)?
7) Related to the previous question: why using confusion learning to detect these thermodynamical properties, instead of conventional Monte Carlo approaches? I suggest you to comment about this in the paper.
8) It is not very clear to me what Figure 1 d) represents precisely. You are considering the two point correlators, thus I suppose you are plotting $f_{c_1}(a_1) \cdot f_{c_2}(a_2)$, where $f$ is the learned convolutional filter and $c_1,c_2$ are the channels $\{\uparrow, \downarrow\}$. Am I right?
9) The notation in equations 6 and 7 is not very clear. In particular, I recommend to define c_1 and c_d independently, to avoid misunderstandings. Also, I believe it is quite standard to use the symbol <<i,j>>, in place of <i,j>_{(d)}, to indicate (diagonal) next-nearest-neighbor sites in the 2D geometry. Moreover, perhaps you can use the overline symbol to indicate the average over a finite batch of snapshots, instead of < > which stands for the full thermodynamic average at a given temperature T. Honestly, I am not understanding what is currently the meaning of the overline in your notation, as for instance in Figure 2 on the y axis labels.
10) In Sec. 2.2.2 (page 9), I am not understanding the sentence: "Correct instances C, in turn, match the distribution A ∪ B/A ∩ B, corresponding to the maximum of the total number of snapshots in A, B for each c1 bin". Could you please clarify this point? What does it mean mathematically "the distribution A ∪ B/A ∩ B"?
11) The main point of Sec. 2.2.2 is to verify that the Neural Network makes predictions on the basis of its knowledge of the full probability distribution of c_1. In particular, you argue that the Network acts by a majority decision. As far as I understood, this means that the Network evaluates how likely it is that c_1 takes the observed value for a state belonging to the class $T<T'$ or to the class $T>T'$, and then it decides by selecting the class for which class there is the highest probability. Am I right? Since this is an important message of your work, I recommend to improve a bit the presentation of this aspect.
12) In Sec. 3 some extra introductory lines about transformers are necessary. I think that most of the physicists (even if with a background in Neural Networks) don't know pretty much nothing about this kind of architecture. In particular, I think you have to provide an idea of how transformers work, why they are supposed to be "able to intrinsically capture long-range correlations", why their use can boost the method in practice (and in which cases).
13) Related to the previous question: I believe you should clarify a bit your claim that: "By combining the confusion learning scheme with transformer neural networks, our work opens new
directions in interpretable quantum image processing being sensible to long-range order."
Author: Henning Schloemer on 2023-06-28 [id 3768]
(in reply to Report 1 on 2023-05-09)
-------------Strengths---------------- 1) The work concerns a very interesting "hot" research topic 2) The numerics performed by the authors seems reliable 3) The authors address very carefully the issue of understanding the mechanisms underlying the goodness of the network predictions. This effort is particularly valuable since in many other works Neural Networks are used as a black box, without any physical interpretation of the learned operating mechanism.
We thank the referee for their time in carefully reviewing our manuscript. We are pleased to hear that the referee commends the relevance of our work, and that they appreciate our efforts in interpreting the network's predictions on physical grounds.
-------------Weaknesses---------------- 1) I think that the paper is not clear enough in some relevant points (see list below), especially concerning the general features and improvements given by their technique, the possibility of using the approach for other physical models and the use of the "transformer" architecture.
We believe that our manuscript greatly increased in clarity and benefited from the referee's comments, which we would like to thank them for.
2) Overall, a very clear/innovative/unambiguous take home message is not clearly recognizable.
Unveiling the microscopic origins of strongly correlated many-body phases is one of the central challenges in modern condensed matter physics. By proposing our neural network approach based on confusion learning and correlator CNNs, we present a tool that can act as a guiding hand when analyzing snapshots of many-body wave functions while having full interpretability in terms of correlation functions. We have adapted the introduction and discussion, to better differentiate our method from existing proposals and hopefully better frame the above mentioned central message.
----------------Report----------------- The paper "Fluctuation based interpretable analysis scheme for quantum many-body snapshots" by Henning Schloemer and Annabelle Bohrdt is a valuable work concerning an extremely relevant field of research, namely the analysis and interpretation of the quantum snapshots acquired via projective measurements from a correlated quantum state. In this paper, the snapshots are analyzed by means of a Correlation Convolution Neural Network, within the framework of Confusion learning. Although these two tools are already known in litterature, the idea of combining them together is original and interesting, leading to an interpretable tool. The authors discuss one main application of their technique, namely the 2D Heisenberg model, for which they argue to be able to identify a qualitative change of the thermodynamic properties at temperature T~0.9J, corresponding to a maximum of the susceptibility. The Authors provide a very detailed interpretation of the learned operating mechanism of the Neural Network. Considering the Strengths/Weaknesses I mentioned and the questions below, I think the work can be published upon some considerable adjustements.*
We are happy to hear the referee’s publication recommendation after the points raised have been addressed. Reprinted below are the requested changes by the referee together with our corresponding replies.
-----------Requested changes-------------- 1) A very generic question about the confusion learning approach: under which conditions one can be confident that "the model is capable of perfectly distinguishing the two phases", as you claim for the case p′=p_c at page 4? As far as I understand, this is crucial in order to observe a W-shape in the accuracy as a function of p′ (or T′).
Having perfect distinction between the two phases by the network is mentioned in our manuscript mainly as an idealized case to illustrate the mechanism of confusion learning. In realistic applications, perfect distinction and hence a completely sharp W-shape is not expected. Nevertheless, even a "washed out" W-shape as also visible in our case suggests that the network is seeing two qualitatively different sets of data separated by a local maximum of the network’s accuracy. We have added a footnote to page 4, hopefully clarifying the matter. For further examples, we refer to the original proposal of confusion learning in [Nieuwenburg et al.; https://www.nature.com/articles/nphys4037].
2) In Section 2.1 you are using N to indicate the maximum order of correlations considered by the network, whereas in Equation 3 N represents the systems size. Maybe you can use a different notation to avoid a misunderstanding.
We thank the referee for pointing out this notation inconsistency, which we have fixed in the revised manuscript.
3) For which reasons are you focusing only on the 2D Heisenberg model? It is just one of the possible applications, or there is a particular motivation for considering it? As far as I understand, you are interested in establishing if the confusion learning approach can detect a qualitative change in the thermodynamical properties (as the one occurring at T=T), although it is not associated with a proper phase transition. Right? Or rather the reason of interest is given by the link with the low energy physics of the Fermi Hubbard model?*
Indeed, both of these motivations are highly relevant. We chose the 2D Heisenberg model as a prototypical, accessible model with interesting magnetic features -- that, being closely related to the Fermi-Hubbard model at half filling, are relevant also in the context of doped Mott insulators. Our results suggest that the model can efficiently detect changes of long-range thermodynamic properties, which in turn paves the way towards novel microscopic insights into the physics of doped Fermi-Hubbard systems. Thus, our given example acts as a proof-of-principle of our co-CCNN network, which we propose as a guiding tool when analyzing snapshots in, for instance, the pseudogap regime of the doped Femi-Hubbard model. In particular, measuring clean snapshots in this regime is possible with current/next generation quantum gas microscopes. We have added the above discussion to the manuscript in Sec. 2.2, clarifying our motivation for studying the Heisenberg model with interpretable machine learning techniques.
4) I believe that before considering one particular application (i.e. before of Sec. 2.2), you should clarify which are the key features of your technique and in which aspects it leads to an improvement with respect to already known techniques.
With the co-CCNN protocol we proposed, interpretable phase classification can be done in a single step (i.e. both classification and interpretation are accessible with a single training run), which greatly simplifies similar multi-step approaches for interpretable phase detection (see e.g. Ref. [11]). We agree with the referee that this point deserves further clarification early on in the text. We have adapted the introduction and section 2.1. of our manuscript, where we highlight the above improvements compared to other techniques.
5) Some other relevant references on the 2D Heisenberg model shall be added, in order to assist a reader that is not fully familiar with the model and its equilibrium properties. Ref.[55],[60] are good, but probably having some more recent works/reviews/books would help...
We have added additional book references (Assa Auerbach, “Interacting Electrons and Quantum Magnetism” and Altland and Simons, "Condensed Matter Field Theory") to the manuscript (before Eq. (2)), where the Heisenberg model and its properties are pedagogically introduced.
6) The agreement between T_max'~0.8 and T~0.9 is not too good. Is there any way of improving the result? Do you expect to find T_max' closer to the peak of specific heat or to the the peak of the susceptibility (i.e. T∗)? *
As we find in our work, the network utilizes the full counting statistics (FCS) of nearest neighbor correlations to make its classification decision. As this crucially depends on the widths of the distributions, we analyzed in more detail the behavior of the fluctuations of c_1, containing contributions from long-range four-point correlations. Though similarities between the FCS measure and thermodynamic quantities such as C_V and chi_s are present, there exists no direct relation. By indirectly evaluating long-range properties (as appearing in $\chi_s$ and $C_V$) of four-point correlations (as appearing in $C_V$), the network succeeds in detecting qualitative changes in the snapshots as a function of temperature, i.e., when spin correlations become significantly long-range. These detected changes cannot, however, directly be attributed to originating from the peak in $C_V$ or $\chi_s$, and shall rather be interpreted as a related but independent indicator of qualitative change close to the pseudogap regime -- as also suggested by the position of the performance maximum lying in between the peaks of $C_V$ and $\chi_s$. Nevertheless, the presence of pronounced signatures of $\sigma_1$ as a function of temperature is very intriguing by itself, in turn strongly encouraging the observation of similar indications at finite doping in spin-resolved occupation number snapshots as accessed through quantum gas microscopes.
We have added the above comment to the manuscript, hopefully clarifying the interpretation of the network’s output and better framing the results.
7) Related to the previous question: why using confusion learning to detect these thermodynamical properties, instead of conventional Monte Carlo approaches? I suggest you to comment about this in the paper.
Though the 2D Heisenberg model and its thermodynamic properties are well understood, there exist many strongly correlated phases where their characterizing features are far less clear (as, for instance, in many of the appearing phases in the Fermi-Hubbard model). Thus, the Heisenberg model acts as a valuable testing ground for our proposed machine learning based classification routine, which can act as a guiding tool towards uncovering microscopic origins of many-body states. The thermodynamic properties presented in Fig. 1 (b) are indeed calculated from Monte Carlo techniques, which the network architecture is shown to pick up upon in an unbiased manner. This, in turn, opens the door to study more complex many-body phases using our protocol.
8) It is not very clear to me what Figure 1 d) represents precisely. You are considering the two point correlators, thus I suppose you are plotting f_c1(a1)f_c2(a2), where f is the learned convolutional filter and c1, c2 are the channels {↑,↓}. Am I right?*
This is precisely what is plotted in Fig. 1 (d), which we have now made explicit in the caption of Fig. 1 as well as in the text in the revised manuscript.
9) The notation in equations 6 and 7 is not very clear. In particular, I recommend to define c_1 and c_d independently, to avoid misunderstandings. Also, I believe it is quite standard to use the symbol <<i,j>>, in place of <i,j>_{(d)}, to indicate (diagonal) next-nearest-neighbor sites in the 2D geometry. Moreover, perhaps you can use the overline symbol to indicate the average over a finite batch of snapshots, instead of < > which stands for the full thermodynamic average at a given temperature T. Honestly, I am not understanding what is currently the meaning of the overline in your notation, as for instance in Figure 2 on the y axis labels.
We agree with the referee that defining c_1 and c_d separately makes it easier to follow the manuscript. Likewise, we acknowledge the referee's suggestion of denoting only full thermodynamic averages by <>, whereas indicating approximations via snapshots by overlined symbols; we have changed the manuscript accordingly.
10) In Sec. 2.2.2 (page 9), I am not understanding the sentence: "Correct instances C, in turn, match the distribution A ∪ B/A ∩ B, corresponding to the maximum of the total number of snapshots in A, B for each c1 bin". Could you please clarify this point? What does it mean mathematically "the distribution A ∪ B/A ∩ B"?
By comparing the distributions in Fig. 3 (a) and (b), one notices that the hatched overlap region in Fig. 3 (b) corresponds to the "wrong instance distribution" of the network's performance in Fig. 3 (a), whereas the non-hatched area is consistent with the "correct instance distribution" of the network. We have adapted the formulation and notation in Sec. 2.2.2, hopefully clarifying the correspondences and notation.
11) The main point of Sec. 2.2.2 is to verify that the Neural Network makes predictions on the basis of its knowledge of the full probability distribution of c_1. In particular, you argue that the Network acts by a majority decision. As far as I understood, this means that the Network evaluates how likely it is that c_1 takes the observed value for a state belonging to the class T<T' or to the class T>T', and then it decides by selecting the class for which class there is the highest probability. Am I right? Since this is an important message of your work, I recommend to improve a bit the presentation of this aspect.
Yes, this is precisely right. By processing the FCS of the {c_1} distributions, the network learns a cutoff c_1', above (below) which it classifies snapshots to belong to T<=T' (T>T'). This learned threshold, in turn, depends explicitly on the means and widths of the distributions, the latter corresponding to the fluctuations of c_1. We have extended the discussion in Sec. 2.2.2 and added a new figure (Fig. 3), putting more emphasis on communicating this central point.
12) In Sec. 3 some extra introductory lines about transformers are necessary. I think that most of the physicists (even if with a background in Neural Networks) don't know pretty much nothing about this kind of architecture. In particular, I think you have to provide an idea of how transformers work, why they are supposed to be "able to intrinsically capture long-range correlations", why their use can boost the method in practice (and in which cases).
We agree with the referee that a broader introduction of transformers is useful in the text. We have extended Sec. 3 in the manuscript, i.p. by explaining in more detail how transformers work, how they can intrinsically capture long-range correlations, and how they can possibly surpass convolutional networks in practice when analyzing snapshots of many-body phases.
13) Related to the previous question: I believe you should clarify a bit your claim that: "By combining the confusion learning scheme with transformer neural networks, our work opens new directions in interpretable quantum image processing being sensible to long-range order."
By applying the all-to-all attention mechanism to many-body snapshots, long-range inter-dependencies between the pixels (i.e. physical sites) can be learned. This, in turn, suggests that long-range as well as non-local order can be captured by transformer networks, possibly being interpretable on physical grounds via the learned attention maps. The expanded discussion on transformer neural networks in Sec. 3 hopefully now justifies our proposal of utilizing the strengths of transformers for quantum-image processing.
Author: Henning Schloemer on 2023-06-28 [id 3771]
(in reply to Report 4 on 2023-05-19)-------------Strengths---------------- 1) The paper addresses an interesting and important research question. In particular, an answer can lead to NNs guiding our understanding of novel phases read straight from experimental snapshots of quantum systems. 2) The way the authors combined interpretability and unsupervised learning within the same tool is very useful and surpasses completely separate procedures for unsupervised phase detection and detection of correlators (done, e.g., in Ref. 11). In other words, the network responsible for unsupervised classification also explains its decision, instead of using separate clustering and then using the identified clusters as labels for supervised interpretable classification. 3) The paper describes nicely most steps of a quite complex procedure, avoiding just citing the previous works. From the ML side, the literature review is thorough.
We thank the referee for their time in carefully reviewing our manuscript. We are pleased to hear that the referee commends the relevance of our work, and that they appreciate our efforts in creating an accessible tool for interpretable phase detection. Reprinted below are the points raised by the referee together with our corresponding replies.
-------------Weaknesses---------------- There are two elements of the paper that would benefit the most from a more thorough explanation: 1) I'm confused by the argument that the NN learns fluctuations of c1/d instead of just c1/d. The explanation needs to be more careful (I will elaborate in "Requested changes"), especially that it was regarded important enough to enter the title.
We find that the correlations the network outputs as important contributions for its decision do not, by themselves, feature any qualitative changes at the temperature of maximum accuracy (see Fig. 2). Instead, the network studies the full counting statistics of nearest neighbor correlations. In particular, these crucially depend on the widths of the distributions, corresponding to the fluctuations of $c_1$. As we present in the manuscript, quantities such as standard deviations $\sigma_1$ as well as four-point correlations as appearing in the widths of $c_1$ show qualitatively different behavior when crossing T’_max, which is why we put emphasis on the term “fluctuations”. In the revised manuscript, we have worked on explaining the above more carefully, hopefully now better communicating this central message.
2) The transformer entering the picture in the end is quite surprising. Justification of including this result should be more thorough (again, see the "Requested changes").
We agree that a more extensive discussion of the transformer section is useful in the text, and refer to “Requested changes” for our reply.
-------------Report---------------- The presented results are an interesting solution to the lack of interpretability of unsupervised classifiers of phases of matter. The paper would, however, benefit from expanding the explanation behind the "fluctuation-based classification" and including results of a classification done with a transformer. After the suggested changes are addressed, I am happy to recommend publication of the paper.
We are happy to hear that the referee recommends our manuscript for publication after corresponding changes have been made. We believe that our manuscript greatly benefited from the referee’s comments, which we would like to thank them for. Below are our replies to the requested changes.
-----------Requested changes-------------- - I suggest elaborating shortly that in this specific case 2x2 filter is enough and that while "the results to not alter qualitatively when choosing larger convolutional windows", there may be systems with longer-range order which would require larger filers and therefore it's a hyperparameter that needs to be validated.
We agree with the referee that it is useful to put more emphasis on the choice of the kernel size in the text, which we have worked on. Additionally, we added a new section to the Appendix, where we compare and discuss the network’s output in more detail for larger filter sizes.
- It would be useful to write in the text that c1 and cd are nearest-neighbor and diagonal correlators (e.g., just before Eq. 6), because c1/d is a bit confusing.
We have worked on clarifying the notation by separately defining $c_1$ and $c_d$, which hopefully makes it easier to follow the manuscript.
- Coming back to the first indicated weakness of the paper. As I understand, you point that the classification cannot be made solely on c1/d as they "show a monotonous increase with decreasing temperature with no qualitative difference above or below the temperature of maximum network accuracy". Why an NN couldn't just learn a threshold for which the phase transition occurs?
Indeed, the network effectively learns a threshold value $c_1'$, above (below) which it classifies a given snapshot as belonging to $T \leq T' (T>T')$. However, the network has no information about the temperatures of the individual snapshots, such that it could not, for instance, choose the mean value of $c_1$ at temperature T' as its threshold. Instead, the network explicitly leverages the full counting statistics of the two snapshot subsets {$T \leq T'$}, {$T>T'$} in order to learn $c_1'$ that maximizes its classification accuracy. In particular, note that the ideal choice for $c_1'$ depends on the widths of the two distributions {$c_1$ | $T \leq T'$}, {$c_1$ | $T>T'$}, which are directly linked to the fluctuations of $c_1$. Hence, an ideal threshold value $c_1’$ can be learned by studying the fluctuations of the correlator $c_1$. In the revised manuscript, we have put more emphasis on communicating this central point, see the extended discussion in Section 2.2.2 and the added Fig. 3.
- Now, the second indicated weakness of the paper. I may know what the authors are hinting at with including the transformer results but it is still not a well-known architecture in a community. I would suggest at least explaining the attention mechanism, show how this can surpass the CNN filter (leading to the ability to "intrinsically capture long-range correlations"), and how it could, in principle, lead to interpretable classification by showing that analysis of elements of attention map would lead to identifying the dominant correlators (even if multiple patches would definitely render the analysis nontrivial).
We agree with the referee that a broader introduction of transformers is useful in the text. We have extended Sec. 3 in the manuscript, i.p. by explaining in more detail how transformers work, how they can intrinsically capture long-range correlations, and how they can possibly surpass convolutional networks in practice when analyzing snapshots of many-body phases. This hopefully communicates the broader picture and outlook more clearly.
An optional comment: - I see that you avoid the boundary effects by cutting 40x40 snapshots to 16x16. However, are the boundary effects playing any role in picking up the correlators by the CCNN? My intuition would be that the dominant correlators should stay the same even after including the boundary?
As the referee suspects, for the case of the Heisenberg model, we do not expect any changes in the network's decision making process. When feeding in snapshots with open boundaries, the quantitative values of $c_1^{(s)}$ are expected to slightly change, however without any qualitative effect on the network's processing procedure.
- As Ref. 11 already combined CCNN with unsupervised techniques, it would be useful to explain why your approach is better (e.g., as I already mentioned in the paper's strengths)
Indeed, as the referee has pointed out as one of the strengths of our paper, one of the main advantages of the co-CCNN scheme is the simple, single-step procedure to perform interpretable phase detection. In the introduction and Sec. 2.1, we have put more emphasis on this point, and thank the referee again for their constructive report.

---

## Round 1 · Referee Report · Anonymous · 2023-5-16

Strengths
- This paper focuses on an important aspect of Machine Learning (ML) applications to classify quantum many-body phases: the physical interpretability of trained neural networks.
- In particular, they propose an ML approach to identify the relevant physical correlations that distinguish two physical regimes. For the specific case treated in the paper, the 2D Heisenberg model, they present solid arguments that their approach identifies the full counting statistics of two-body correlations as the main element for classifying low- and high-temperature regimes.
Weaknesses
- The paper could have discussed the feasibility of applying the proposed approach for more generic cases. For example, if one of the physical regimes is characterized by higher-order correlations (going beyond 4-body terms), the approach can still be efficiently applied.
In particular, what is the computational cost of considering larger CNN filters?
- I would expect at least a discussion of how the accuracy of their results (e.g., the estimation of the decision boundary T’) is affected by changing hyperparameters of the CCNN, for example, the CNN filter.
Report
The paper proposes a Machine Learning approach to classify phases of matter based on correlation convolution neural network (CCNN) and confusion learning. In particular, they show that their approach can distinguish two physical regimes and detect the dominant pattern of correlations that distinguishes such regimes.
Overall, the paper is well-written and makes progress in identifying strategies to perform interpretable Machine Learning.
After addressing the comments raised here (Weakness and Requested changes), I think this paper can be published in SciPost.
Requested changes
(1) Compare the four-point correlations shown in Fig. 4 (b) with two-point correlations. Is there any advantage in considering the 4-point correlator (instead of 2-point correlations) to distinguish the two thermal regimes 2D Heisenberg model?
(2) Is it possible to rule out if higher-order correlations (going beyond 4-point correlations) play any role in distinguishing the two thermal regimes?
It would be important to address these questions.
Author: Henning Schloemer on 2023-06-28 [id 3769]
(in reply to Report 2 on 2023-05-16)
-------------Strengths---------------- - This paper focuses on an important aspect of Machine Learning (ML) applications to classify quantum many-body phases: the physical interpretability of trained neural networks. - In particular, they propose an ML approach to identify the relevant physical correlations that distinguish two physical regimes. For the specific case treated in the paper, the 2D Heisenberg model, they present solid arguments that their approach identifies the full counting statistics of two-body correlations as the main element for classifying low- and high-temperature regimes.
We thank the referee for their time in carefully reviewing our manuscript. We are pleased to hear that the referee commends the relevance of our work, and that they appreciate our efforts in interpreting the network's predictions on physical grounds.
-------------Weaknesses---------------- - The paper could have discussed the feasibility of applying the proposed approach for more generic cases. For example, if one of the physical regimes is characterized by higher-order correlations (going beyond 4-body terms), the approach can still be efficiently applied. In particular, what is the computational cost of considering larger CNN filters? - I would expect at least a discussion of how the accuracy of their results (e.g., the estimation of the decision boundary T’) is affected by changing hyperparameters of the CCNN, for example, the CNN filter.
We agree that a more detailed discussion of results when changing the hyperparameters of the network is a useful addition to our analysis. We have added a new section to the Appendix, where we analyze in more detail the network’s output when choosing larger convolutional windows of size 3x3. We further clarified in the main text that the filter size shall be treated as a tunable hyperparameter, and must be chosen with care in particular if higher-order correlations characterize a given regime.
As in standard CNN architectures, numerical costs scale quadratically with the linear filter size. However, as the data sets and image sizes of the analyzed snapshots are comparably small (2000 snapshots per temperature, 16x16 images), the training for all decision boundaries is computationally feasible and can be done with very low computational resources even for larger kernel sizes. We note that in experimentally realistic settings, similar image and data set sizes are expected, rendering the approach practical for quantum gas image applications.
-------------Report---------------- The paper proposes a Machine Learning approach to classify phases of matter based on correlation convolution neural network (CCNN) and confusion learning. In particular, they show that their approach can distinguish two physical regimes and detect the dominant pattern of correlations that distinguishes such regimes. Overall, the paper is well-written and makes progress in identifying strategies to perform interpretable Machine Learning. After addressing the comments raised here (Weakness and Requested changes), I think this paper can be published in SciPost.
We are happy to hear the positive feedback by the referee and their recommendation for publication once the comments raised have been addressed. We believe that our manuscript greatly benefited from the referee’s comments, which we would like to thank them for. Reprinted below are the requested changes raised by the referee together with our corresponding replies.
-----------Requested changes-------------- (1) Compare the four-point correlations shown in Fig. 4 (b) with two-point correlations. Is there any advantage in considering the 4-point correlator (instead of 2-point correlations) to distinguish the two thermal regimes 2D Heisenberg model?
Indeed, in the (nearest-neighbor) two-point correlations, no signals indicating a change of thermodynamic properties are found in the Heisenberg model, as shown in Fig. 2 of the manuscript. By using the co-CCNN scheme, we find that the network utilizes the full counting statistics of nearest neighbor correlations – which directly include information about the widths of the distributions. These widths, in turn, include contributions from four-point correlations of two nearest neighbor pairs, which as we show do yield signals of qualitative change of the thermodynamic properties. Hence, with access to only local correlations within the convolutional window, the network finds a way to assess long-range properties of the system by evaluating the full distribution of nearest neighbor correlations. Note that, when considering long-range two-point correlations, a qualitative change is visible via a peak in the susceptibility, which, however, the network has no access to due to its spatial constraint to analyze correlations only within the convolutional window. We have put more emphasis on this in the revised manuscript, hopefully clarifying this central point.
(2) Is it possible to rule out if higher-order correlations (going beyond 4-point correlations) play any role in distinguishing the two thermal regimes?
Without further knowledge of the system, it can not generally be ruled out that higher-order correlations also show signals of distinguishing the two thermal regimes. Nevertheless, we note that the maximal order of correlations the network analyzes is a tunable hyperparameter of the system (i.e. N_max, cf. Eq. (1) and Fig. 1 (a)). Upon changing the size of the filter and N_max (e.g. with filter sizes 3x3 up to 9th order correlations are analyzed by the network), our results stay qualitatively the same – i.e., two-point correlations single out as the important quantities for the network, which is highly indicative that local, higher order correlations play no significant role in distinguishing the thermal regimes. We have noted and underlined the above in the revised manuscript, and have added a new section to the Appendix where we analyze the network’s output using 3x3 filters in more detail.

---

## Round 1 · Referee Report · Anonymous · 2023-5-19

Strengths
1) The paper addresses an interesting and important research question. In particular, an answer can lead to NNs guiding our understanding of novel phases read straight from experimental snapshots of quantum systems.
2) The way the authors combined interpretability and unsupervised learning within the same tool is very useful and surpasses completely separate procedures for unsupervised phase detection and detection of correlators (done, e.g., in Ref. 11). In other words, the network responsible for unsupervised classification also explains its decision, instead of using separate clustering and then using the identified clusters as labels for supervised interpretable classification.
3) The paper describes nicely most steps of a quite complex procedure, avoiding just citing the previous works. From the ML side, the literature review is thorough.
Weaknesses
There are two elements of the paper that would benefit the most from a more thorough explanation:
1) I'm confused by the argument that the NN learns fluctuations of $c_{1/d}$ instead of just $c_{1/d}$. The explanation needs to be more careful (I will elaborate in "Requested changes"), especially that it was regarded important enough to enter the title.
2) The transformer entering the picture in the end is quite surprising. Justification of including this result should be more thorough (again, see the "Requested changes").
Report
The presented results are an interesting solution to the lack of interpretability of unsupervised classifiers of phases of matter. The paper would, however, benefit from expanding the explanation behind the "fluctuation-based classification" and including results of a classification done with a transformer. After the suggested changes are addressed, I am happy to recommend publication of the paper.
Requested changes
Strongly recommended changes:
- I suggest elaborating shortly that in this specific case 2x2 filter is enough and that while "the results to not alter qualitatively when choosing larger convolutional windows", there may be systems with longer-range order which would require larger filers and therefore it's a hyperparameter that needs to be validated.
- It would be useful to write in the text that $c_1$ and $c_d$ are nearest-neighbor and diagonal correlators (e.g., just before Eq. 6), because $c_{1/d}$ is a bit confusing.
- Coming back to the first indicated weakness of the paper. As I understand, you point that the classification cannot be made solely on $c_{1/d}$ as they "show a monotonous increase with decreasing temperature with no qualitative difference above or below the temperature of maximum network accuracy". Why an NN couldn't just learn a threshold for which the phase transition occurs?
- Now, the second indicated weakness of the paper. I may know what the authors are hinting at with including the transformer results but it is still not a well-known architecture in a community. I would suggest at least explaining the attention mechanism, show how this can surpass the CNN filter (leading to the ability to "intrinsically capture long-range correlations"), and how it could, in principle, lead to interpretable classification by showing that analysis of elements of attention map would lead to identifying the dominant correlators (even if multiple patches would definitely render the analysis nontrivial).
An optional comment:
- I see that you avoid the boundary effects by cutting 40x40 snapshots to 16x16. However, are the boundary effects playing any role in picking up the correlators by the CCNN? My intuition would be that the dominant correlators should stay the same even after including the boundary?
- As Ref. 11 already combined CCNN with unsupervised techniques, it would be useful to explain why your approach is better (e.g., as I already mentioned in the paper's strengths)

---

## Round 1 · Referee Report · Everard van Nieuwenburg · 2023-5-19

Strengths
1 - Clearly written
2 - Good mix between method and application to physics
3 - Nice demonstration of physical feature identification by combining two known methods
Weaknesses
1 - More 'correlation' based than 'fluctuation' based
2 - Not really clear why, for the Heisenberg model, the 'broad match' of the network model is considered useful from the text
3 - Transformer 'bonus' feels slightly out of place, in the sense that the authors report on its performance (and T_c ~ 0.6 feature), but leave interpretation of that to future work.
Report
This work integrates two established methods in an attempt to create a more interpretable model for phase transition detection & analysis based on physical snapshot-like data. Namely, the authors use the confusion method (and explain it well) and correlator-based convolutional networks (and explain those well, too). The latter allows for more interpretable results based on 'which correlation maps are important in the classification'-type analysis (using a regularization path analysis), and the former is what is used for the overall training routine.
I particularly enjoyed the authors' insight on the linear slopes in the network's performance when trained using the confusion method:
"we see a linear reduction of accuracy, signaling that the network makes a majority decision".
The authors focus on application to the Heisenberg model, and make a good case that the correlation maps the model learns are the nearest-neighbour two-point correlators. That leads to an interesting analysis on the full-counting statistics (FCS), and the authors convinced me that indeed the FCS of the 'c1' correlator plausibly is what allows for the classification. What I think is missing here a little, would be a reflection on how this could lead to the broad peak of the performance that 'averages' between the features of C_v and X_s? Is there a way to identify the contributions of each to the model?
Could the authors say a little more about the computational cost for this approach? I wonder specifically about the re-training in the confusion method, and would want to point the authors to a more automated version we named "Discriminative Cooperative Networks".
Requested changes
1 - A short reflection on the feasibility of distinguishing the C_v and X_s contributions to the network model, to argue that the model's performance peak does not indicate a transition directly.
Author: Henning Schloemer on 2023-06-28 [id 3770]
(in reply to Report 3 by Everard van Nieuwenburg on 2023-05-19)
-------------Strengths----------------
*1 - Clearly written
2 - Good mix between method and application to physics
3 - Nice demonstration of physical feature identification by combining two known methods*
We thank the referee for their time in carefully reviewing our manuscript and are happy to hear that the referee commends the demonstration of our work. Reprinted below are the points raised by the referee together with our corresponding replies.
-------------Weaknesses----------------
*1 - More 'correlation' based than 'fluctuation' based*
We find that the correlations the network outputs as important contributions for its decision do not, by themselves, feature any qualitative changes at the temperature of maximum accuracy (see Fig. 2). Instead, the network studies the full counting statistics of nearest neighbor correlations. In particular, these crucially depend on the widths of the distributions, corresponding to the fluctuations of $c_1$. As we present in the manuscript, quantities such as standard deviations $\sigma_1$ as well as four-point correlations as appearing in the widths of $c_1$ show qualitatively different behavior when crossing T’_max, which is why we put emphasis on the term “fluctuations”.
*2 - Not really clear why, for the Heisenberg model, the 'broad match' of the network model is considered useful from the text*
The broad match between the maximum of the network’s performance and the maxima of the susceptibility and specific heat suggest that the network picks up upon qualitative changes of thermodynamic properties of the Heisenberg model, where significant long-range order setting in below a certain temperature strongly influences the behavior of both $C_V$ and $\chi_s$. We have emphasized this in the revised manuscript, hopefully clarifying the matter.
*3 - Transformer 'bonus' feels slightly out of place, in the sense that the authors report on its performance (and T_c ~ 0.6 feature), but leave interpretation of that to future work.*
We agree that a more extensive discussion of the transformer section is useful in the text. We have extended Sec. 3 in the manuscript, i.p. by explaining in more detail how transformers work, how they can intrinsically capture long-range correlations, and how they can possibly surpass convolutional networks in practice when analyzing snapshots of many-body phases – thus framing the confusion transformer section in a more complete manner.
-------------Report----------------
*This work integrates two established methods in an attempt to create a more interpretable model for phase transition detection & analysis based on physical snapshot-like data. Namely, the authors use the confusion method (and explain it well) and correlator-based convolutional networks (and explain those well, too). The latter allows for more interpretable results based on 'which correlation maps are important in the classification'-type analysis (using a regularization path analysis), and the former is what is used for the overall training routine.
I particularly enjoyed the authors' insight on the linear slopes in the network's performance when trained using the confusion method: "we see a linear reduction of accuracy, signaling that the network makes a majority decision".
The authors focus on application to the Heisenberg model, and make a good case that the correlation maps the model learns are the nearest-neighbour two-point correlators. That leads to an interesting analysis on the full-counting statistics (FCS), and the authors convinced me that indeed the FCS of the 'c1' correlator plausibly is what allows for the classification.*
We are happy to hear the positive feedback by the referee, and that they appreciate our efforts in interpreting the network’s decision process. We believe that our manuscript greatly benefited from the referee’s comments, which we would like to thank them for.
*What I think is missing here a little, would be a reflection on how this could lead to the broad peak of the performance that 'averages' between the features of C_v and X_s? Is there a way to identify the contributions of each to the model?*
We refer to the reply in “Requested changes” below.
*Could the authors say a little more about the computational cost for this approach? I wonder specifically about the re-training in the confusion method, and would want to point the authors to a more automated version we named "Discriminative Cooperative Networks".*
As the data sets and image sizes of the analyzed snapshots are comparably small (2000 snapshots per temperature, 16x16 images), the retraining for all decision boundaries is computationally feasible and can be done with very low computational resources. We note that in experimentally realistic settings, similar image and data set sizes are expected, rendering the approach practical for quantum gas image applications. Nevertheless, the automated DCNs the referee and their co-author suggested is very interesting and relevant also for our purposes, possibly opening future directions of research by combining the CCNN architecture with guesser and learner networks. We have included a short discussion of the above in the revised manuscript in the discussion section.
-----------Requested changes--------------
*1 - A short reflection on the feasibility of distinguishing the C_v and X_s contributions to the network model, to argue that the model's performance peak does not indicate a transition directly.*
Though similarities between the FCS measure that the network utilizes and thermodynamic quantities such as $C_V$ and $\chi_s$ are present, there exists no direct relation. By having access to the distribution widths, the network indirectly evaluates long-range properties (as appearing in $\chi_s$ and $C_V$) of four-point correlations (as appearing in $C_V$), which enables it to detect qualitative changes in the snapshots as a function of temperature. These detected changes cannot, however, directly be attributed to originating from the peak in $C_V$ or $\chi_s$, and shall rather be interpreted as a related but independent indicator of qualitative change close to the pseudogap regime -- as also suggested by the position of the performance maximum lying in between the peaks of $C_V$ and $\chi_s$. Nevertheless, the presence of pronounced signatures of $\sigma_1$ as a function of temperature is very intriguing by itself, in turn strongly encouraging the observation of similar indications at finite doping in spin-resolved occupation number snapshots as accessed through quantum gas microscopes.
We have added the above comment to the manuscript, hopefully clarifying the interpretation of the network’s output and better framing the results.

---

## Round 2 · Referee Report · Everard van Nieuwenburg (Referee 3) · 2023-7-6

Report

This revised version of the manuscript is a clear improvement over v1, in my opinion. The message of the paper is more pronounced, and previously stated but unmotivated parts of the text now have proper context.

I remain positive in my assessment. Especially taken together with the other reports on v1, and the authors' implementation of that feedback, I have no further requested changes.

---

## Round 2 · Referee Report · Anonymous (Referee 2) · 2023-7-8

Report

The authors satisfactorily addressed my comments and suggestions and improved the new version of the manuscript. Therefore, I am happy to recommend the paper for publication in SciPost.

---

## Round 2 · Referee Report · Anonymous (Referee 1) · 2023-7-15

Report

The authors have taken my comments and suggestions into consideration and made significant improvements to the latest version of the manuscript. As a result, I am pleased to endorse the paper for publication in SciPost.

---

## Editorial Decision

published